# Modeling In Vitro Osteoarthritis Phenotypes in a Vascularized Bone Model Based on a Bone-Marrow Derived Mesenchymal Cell Line and Endothelial Cells

**DOI:** 10.3390/ijms22179581

**Published:** 2021-09-03

**Authors:** Alessandro Pirosa, Esma Bahar Tankus, Andrea Mainardi, Paola Occhetta, Laura Dönges, Cornelia Baum, Marco Rasponi, Ivan Martin, Andrea Barbero

**Affiliations:** 1Department of Biomedicine, University Hospital Basel, University of Basel, 4056 Basel, Switzerland; alessandro.pirosa@unibas.ch (A.P.); esma.tankus@unibas.ch (E.B.T.); andrea.mainardi@usb.ch (A.M.); laura.doenges@usb.ch (L.D.); ivan.martin@usb.ch (I.M.); 2Department of Electronics, Information and Bioengineering, Politecnico di Milano, 20133 Milan, Italy; paola.occhetta@polimi.it (P.O.); marco.rasponi@polimi.it (M.R.); 3Department of Biomedical Engineering, University of Basel, 4123 Allschwil, Switzerland; 4Department of Orthopaedic Surgery and Traumatology, University Hospital Basel, 4031 Basel, Switzerland; cornelia.baum@usb.ch; 5Department of Research and Development, Schulthess Klinik Zurich, 8008 Zurich, Switzerland

**Keywords:** osteoarthritis, mesenchymal stromal cells, endothelial cells, inflammation, photocrosslinked, gelatin

## Abstract

The subchondral bone and its associated vasculature play an important role in the onset of osteoarthritis (OA). Integration of different aspects of the OA environment into multi-cellular and complex human, in vitro models is therefore needed to properly represent the pathology. In this study, we exploited a mesenchymal stromal cell line/endothelial cell co-culture to produce an in vitro human model of vascularized osteogenic tissue. A cocktail of inflammatory cytokines, or conditioned medium from mechanically-induced OA engineered microcartilage, was administered to this vascularized bone model to mimic the inflamed OA environment, hypothesizing that these treatments could induce the onset of specific pathological traits. Exposure to the inflammatory factors led to increased network formation by endothelial cells, reminiscent of the abnormal angiogenesis found in OA subchondral bone, demineralization of the constructs, and increased collagen production, signs of OA related bone sclerosis. Furthermore, inflammation led to augmented expression of osteogenic (alkaline phosphatase (ALP) and osteocalcin (OCN)) and angiogenic (vascular endothelial growth factor (VEGF)) genes. The treatment, with a conditioned medium from the mechanically-induced OA engineered microcartilage, also caused increased demineralization and expression of ALP, OCN, ADAMTS5, and VEGF; however, changes in network formation by endothelial cells were not observed in this second case, suggesting a possible different mechanism of action in inducing OA-like phenotypes. We propose that this vascularized bone model could represent a first step for the in vitro study of bone changes under OA mimicking conditions and possibly serve as a tool in testing anti-OA drugs.

## 1. Introduction

The aging population has led osteoarthritis (OA) to become the most prevalent degenerative and disability-causing joint disease worldwide, with a huge burden on economics and social welfare [1]. However, despite OA prevalence, research has made insufficient progress on the development of disease modifying therapies. In fact, present anti-OA pharmacological treatments still rely exclusively on palliatives aimed at relieving pain through the use of non-steroidal and corticosteroid anti-inflammatory drugs [2]. This lack of treatments correlates, at least in part, with the absence of representative models to better understand the pathological mechanisms. Traditionally, progressive degeneration and loss of cartilage were considered as the sole clinical OA hallmarks [3]. However, subchondral bone is also heavily affected in OA, showing sclerosis (thickening and demineralization), osteophyte formation, and abnormal vascularization [4]. Moreover, blood vessels invade the otherwise avascular cartilage and the whole OA joint is characterized by a state of low-grade inflammation with the production of IL-1β, TNFα, and IL-6 by cartilage, but also by the thickened and inflamed synovium [5]. The tight interaction between bone and cartilage, coupled with the chronic inflamed environment, renders it extremely complex to identify the molecular triggers of early stages of OA [6], and there is still no consensus in the scientific community concerning OA origin [7].

During OA, bone undergoes structural and molecular changes. It acquires a sclerotic phenotype: the bone to void ratio increases, but the increased matrix turnover leads to ECM hypo-mineralization [8,9]. Molecular changes also include increased production of osteocalcin, alkaline phosphatase, VEGF, and an increased catabolic activity mediated by aggrecanases (ADAMTS4 and 5) and metalloproteinases (MMP5 and 13) [10,11,12]. Another readily observable phenotype is the increased angiogenesis and the penetration of vasculature into the above lying cartilage, which should not be vascularized in healthy conditions [13]. Overall, all joint tissues are inflamed and characterized by a catabolic environment consisting of overexpressed aggrecanases [14].

Different animal models have been adopted to study the onset of OA. Mice studies evidenced that bone might be implicated in the early stages of OA, due to its mechanical dysregulation in response to overuse or an acute event that, in turn, drives cartilage subsequent degeneration through the production of aggrecanases and metalloproteinases (MMPs) [15]. Moreover, it was observed that, before any cartilage damage is visible, mice pre-osteoclasts produce higher amounts of platelet-derived growth factor BB (PDGF-BB), driving angiogenesis dysregulation in the subchondral bone. This causes, in turn, an unbalanced interaction with the surrounding cartilage, leading to increased production of MMP13 and cartilage degeneration [16]. Different signaling pathways have been studied as potential triggers of OA in the osseous compartment, showing evidence of Wnt activation in subchondral bone and osteophytes of OA mice [12], increased phosphorylation of AKT in mouse models of post-traumatic OA [11], as well as disruption of TGF-β/BMP signaling in OA osteoblasts [10].

Although in vivo models give some insights into the role of bone in OA, they inherently differ from human physiology and do not allow to dissect the different contributions from the diverse tissues of the joint because they lack standardization outcomes [17]. As a result, a long list of failures in OA drugs development can be ascribed to non-anticipated adverse effects and non-homogeneous outcomes in animal studies [18].

Reliable multi-cellular human in vitro models mimicking the diverse tissues involved in OA (i.e., cartilage, but also synovium, bone, and vasculature) could be of paramount importance to tackle their single contributions in order to have a clear understanding of the processes governing the pathology in its early stages. However, currently available in vitro models of OA, from the 2D culture of chondrocyte, to co-culture with synovial cells, to more advanced, mechanically active, microfluidic-based 3D tissue-on-chip settings [19,20], focused mostly on the cartilage compartment.

Notwithstanding, bioreactor technologies allowed the achievement of OA bone models with the development of meso-scale structures, ranging from osteochondral constructs [21,22] to more complex vascularized bone and cartilage interphase systems [23], or even models comprising bone, cartilage, and synovium [24]. Although these systems contain different cellular components reflecting the various OA affected compartments, they require specific bioreactors and scaffold manufacturing technologies, making their widespread adoption more difficult. Moreover, the majority of these complex models rely on the use of primary adult stem/stromal cells, whose performance is highly affected by donor-to-donor variability [25]. The alternative use of induced pluripotent stem cells (iPSCs) is associated with the challenges related to retention of residual epigenetic memory from the cell source, which can lead to potentially biased differentiation towards a specific cell type, accumulation of chromosomal and genetic aberrations, and immature functional characteristics due to their embryonic or fetal origin [26].

To date, we believe that this is the first work to report a simple, highly reproducible in vitro system comprising osseous and vascular components to study OA. For this reason, we developed a model of vascularized bone based on a mesenchymal stromal cell line previously developed in our lab [27]. These cells, from now on referred to as Mesenchymal cells Sword of Damocles (MSODs), were shown to be able to osteogenically differentiate in vitro and form bone once implanted in vivo [28]. MSODs were co-cultured with HUVECs in a photocrosslinkable hydrogel to introduce a vascular component, due to their ability to re-organize in capillary-like structures within that matrix. Several studies have already employed the established method of encapsulating endothelial cells in 3D hydrogels with tissue-specific cells, such as mesenchymal cells [29], neural cells [30], and hepatocytes [31], to generate engineered vascularized tissues to study angiogenesis in bone or the effect of vascularization in neural or hepatocytic function.

The vascularized model proposed here was subjected to different OA-inducing stimuli: (i) an inflammatory cocktail composed of IL-6, IL-1β, and TNFα or (ii) conditioned medium generated from engineered OA cartilage-on-chip subjected to hyperphysiological compression (HPC microcartilage) [20]. We hypothesized that inflammation and/or the factors produced and released by mechanically-induced OA cartilage will lead to OA characteristic changes in the osteoblastic and vascular components of the model, therefore validating and proposing the system as a simple but biologically-relevant model of OA bone for future research in drug target discovery and screening.

## 2. Results

### 2.1. Generation of 3D Constructs Made of MSODs and HUVECs Co-Cultured in gelMA

The model presented in this study was generated by co-culturing an equal ratio of MSODs and HUVECs embedded in photocrosslinked gelatin methacrylate (gelMA). Constructs were exposed to a 1:1 (*v*/*v*) combination of osteogenic medium and endothelial medium for 2 weeks to allow concomitant osteogenesis and tubulogenesis. The differentiation phase was followed by a further week of OA induction through supplementation of inflammatory cytokines or conditioned medium from hyper-physiologically compressed (HPC) microcartilage. MSODs or HUVECs alone were used as controls (Figure 1A).

The constructs’ cell viability was evaluated indirectly through a glucose consumption assay. A decrease of glucose concentration in the medium was registered from day 7 to day 14 for MSODs and MSOD-HUVECs, while the concentration of the metabolite remained unchanged in HUVECs alone. Results were confirmed by DNA quantification, showing an increase from day 7 to day 14 for all conditions except HUVECs alone (Figure 1B). Figure 1C shows the morphology of cultures at day 14, highlighting the uniform distribution of GFP positive MSODs in the constructs and network formation by RFP-positive HUVECs. Positive Calcein Blue AM staining confirmed the viability of cells throughout the culture period.

### 2.2. HUVECs Enhanced Osteogenic Differentiation of MSODs in the Co-Culture System

After verifying viability and morphological robustness, our system was characterized for osteogenic differentiation. HUVECs alone did not show any sign of osteogenesis (Appendix A), thus the extent of mineralization/osteogenesis was mainly assessed in the MSODs and MSOD-HUVECs experimental groups (Figure 2). Alizarin red staining showed a progressive accumulation and uniform distribution of calcium deposits from day 7 to day 14 for both groups (Figure 2A). Quantification of the staining intensity after digesting the constructs revealed higher calcium levels in the MSOD-HUVEC compared to MSODs (Figure 2A).

Immunofluorescence staining showed positive signals for collagen type 1 (COL1A1) and bone sialoprotein (BSP) for both conditions at day 7, whereas Col1 and BSP signals appeared more intense at day 14 for the MSOD-HUVECs group compared to controls (Figure 2B). RT-qPCR analyses showed that the selected early and late osteogenic genes [32] were expressed at similarly low levels in MSODs and MSOD-HUVECs at day 7. Prolonging the culture time, upregulation of the osteogenic genes occurred mainly in the MSOD-HUVEC group, so that, at day 14, all the osteogenic genes were highly expressed in the MSOD-HUVEC (vs. MSOD) group. In particular, at this later time point, statistically significant higher mRNA expression levels of alkaline phosphatase (ALP, 10-fold), osterix (OSX, 80-fold), bone-sialoprotein (BSP, 6-fold), osteopontin (OPN, 2-fold), and osteocalcin (OSC, 2-fold) were observed in MSOD-HUVECs compared to MSODs alone. mRNA expression of COL1A1 remained similar in the two groups (Figure 2C).

Culture media were additionally analyzed to quantify the amounts of ALP activity and calcium released. ALP activity did not change over the culture period in both conditions. Instead, calcium released in the medium statistically decreased from day 7 to day 14 in both conditions, indicating calcium incorporation in the matrix [33] (Figure 2D).

### 2.3. MSOD Helped Stabilization of the HUVEC Network Formation after 2 Weeks of Culture

The constructs were then investigated for the ability of HUVECs to reorganize themselves into networks. HUVECs’ overall morphology and network forming capacity could be assessed given the use of RFP cells. At day 7, HUVECs exhibited networks with similar features in term of branching and vessel density in the two groups. At day 14, instead, we observed a statistically significant reduction in the network formation by HUVECs in monoculture, whereas the co-culture maintained the same as HUVECs’ branching and vessel density at day 7 (Figure 3A). This indicated that MSODs appear to have a role in stabilizing network structures made by HUVECs, similar to the pericyte function exerted by specific subpopulations of MSCs in maintaining endothelial tubular networks when co-cultured with HUVECs [34]. Immunofluorescence pictures showed the presence of HUVECs and osteo-differentiating MSODs through the expression of CD31 and BSP, respectively, at both timepoints (Figure 3B), confirming also the decreased network formation by HUVECs in the monoculture group.

### 2.4. OA Phenotype Induction though Administration of an Inflammatory Cocktail

The next step after evaluating the performance of our model in terms of osteogenesis and network formation was to challenge it with an inflammatory insult to mimic the environment of OA bone by using a cocktail of IL-6, IL-1β, and TNFα at concentrations that can be found in joints of OA patients [35]. A further week of culture in the presence of inflammatory factors induced MSOD-HUVEC constructs to accumulate higher amounts of COL1A1 protein, while we did not see qualitative changes in BSP staining (Figure 4A), to lose a higher extent of the accumulated calcium deposits in terms of decreased alizarin red staining intensity in the constructs (2-fold) and to gain a higher release of calcium in the medium (6-fold), all of which represent phenotypes of subchondral bone sclerosis during OA [9,36]. We also detected a 2-fold enhanced ALP activity in the supernatant of the MSOD-HUVEC compared to the control (Figure 4B), which is another characteristic of subchondral bone during OA [37].

Additional OA traits acquired by the MSOD-HUVEC in response to inflammation consisted of enhanced gene expression of VEGF (2-fold), which has been associated to OA severity [38], COL1A1 (4-fold), and the COL1A1/COL1A2 ratio (10-fold), symptoms of bone sclerosis during OA [9], ADAMTS5 (2-fold) and MMP13 (2-fold), typical catabolic markers in an osteoarthritic environment [39], OSC (3-fold), and ALP (5-fold), identified as downstream targets of the Wnt pathway alteration during OA [12], as well as decreased expression of WNT5A mRNA (0.5-fold) [12] (Figure 4C). In the control group, none of the osteogenic markers (at protein and gene level) were significantly modulated by the exposure to inflammatory factors. We also compared the genes’ expression of the engineered constructs with that of native human sclerotic bone isolated from OA patients who underwent total joint arthroplasty (dashed lines of Figure 4 and Appendix A). Interestingly, most of the genes correlated with the actual expression of human OA bone, in particular COL1A1, COL1A1/COL1A2, OSC, ADAMTS5, and WNT5a, showed very similar expression in both engineered MSOD-HUVEC constructs and native OA bone. We even obtained higher levels of MMP13 mRNA expression in the inflamed constructs compared to the levels found in OA sclerotic bone. These findings indicate that our co-culture system could capture actual phenotypes that characterize native human OA bone. Finally, MSOD-HUVEC constructs exhibited increased network formation in response to inflammation, as shown by enhanced vessel branching (2-fold) and vessel density (2-fold) (Figure 4D,E). Interestingly, HUVEC network formation was not detectable at this timepoint, regardless of the inflammation, indicating a progressive degeneration of networks without the stabilizing presence of the MSOD.

### 2.5. Effects of Conditioned Medium from OA Cartilage-on-Chip

In the joint environment, the subchondral bone is not only exposed to inflammation but also to factors released by other surrounding tissues, e.g., articular cartilage. As an additional characterization of our model, its response upon exposure to the secretome of engineered OA cartilage-on-chip, subjected to hyperphysiological compression (HPC microcartilage), was assessed. In particular, this stimulation was conducted using the conditioned medium from a cartilage-on-chip model exposed to 30% confined compression, recapitulating the mechanical factors involved in OA pathogenesis, previously demonstrated to be sufficient to induce OA traits [20]. Briefly, primary chondrocytes were cultured statically in microfluidic devices for 2 weeks and subjected to hyperphysiological (HPC) loading for 1 additional week, as previously demonstrated [20]. Cartilaginous controls cultured statically for 3 weeks were adopted as controls. Culture medium was collected during the mechanical loading phase and adopted to condition the vascular bone model. Our results showed a decrease in DNA contents in MSOD-HUVEC and HUVEC but not in MSOD constructs upon exposure to the conditioned medium of HPC rather than the statically cultured cartilage (Figure 5A).

Supernatant analysis (Figure 5B) revealed an increased release of calcium from both MSOD and MSOD-HUVEC following exposure to the HPC- (but not static-) conditioned medium, confirming a certain extent of demineralization provoked by OA HPC microcartilage, similar to that induced by the inflammatory cytokines, whereas ALP activity did not change due to treatments in every condition. RT-qPCR analyses (Figure 5C) showed an increase in the mRNA expression of ALP (2-fold), OSC (10-fold), and VEGF (3-fold) in MSOD in response to HPC medium. Instead, in the MSOD-HUVEC, all the analyzed genes remained unchanged following the exposure to both conditioned media. The ability of HUVEC to form networks was not affected by conditioned medium treatment, in part confirming the results of the gene expression, in which this treatment mostly affected just the mesenchymal component of the culture and not the endothelial component (Figure 5D). Overall, these results suggest a potential mechanism of action of the HPC-conditioned medium, different from the mechanism mediated by inflammation in inducing OA traits on our vascularized bone model.

This hypothesis was corroborated by the fact that we did not detect, in the conditioned media from cartilage-on-chip constructs, the presence of previously used inflammatory cytokines, such as IL-1β or TNFα. A slight amount of IL-6 was indeed detectable, yet in negligible concentration if compared to the levels used in the inflammatory condition (Appendix A). We also investigated the conditioned media for presence of IL-8, another cartilage-derived mediator of OA [40], which was found upregulated at the gene level in hyperphysiologically-loaded cartilage-on-chip [20]. Higher IL8 levels were detected in conditioned media from HPC constructs, with respect to static ones, but were still in negligible amounts (<1 pg/mL), with respect to cytokine levels adopted in our inflammatory conditioning.

## 3. Discussion

Our research study focused on the development of a cell line based in vitro vascularized bone construct in which OA traits were induced through inflammatory stimuli or the conditioned medium from OA engineered cartilage. In particular, inflammation induced alterations in both the bony and vascular components of the co-culture model, while the OA cartilage conditioned medium mainly affected the mesenchymal component when endothelial cells were absent in the system. This indicates a crucial role of vasculature in establishing a reliable in vitro inflammation-mediated OA system with the phenotypes present in vivo.

Healthy subchondral bone is characterized by an osteoblast-produced, extracellular matrix, rich in collagen type 1 and hydroxyapatite, which confers mechanical strength, as well as by a vascular network that provides nutrients/waste exchange [41]. Therefore, a critical point in modeling bone in vitro is to introduce at least these two main components. Different strategies followed this direction, exploiting various co-culture systems. Typically, mesenchymal stomal cells were pre-differentiated into osteoblasts and then combined with HUVECs and scaffolds, subsequently implanted in vivo to verify bone formation and vascularization [42,43]. However, this approach has the inherent limitation of using primary cells, such as MSCs, which have been deemed as a heterogeneous cell source subjected to donor-donor and even inter-donor preparation variability [25]. Hence, this study employed a mesenchymal stromal cell line, MSOD, previously developed by our lab [27], which is not hampered by donor and isolation protocol variability and has been shown to differentiate toward osteogenesis in vitro and in vivo [28]. Moreover, MSODs and HUVECs were co-cultured from the beginning, without any pre-differentiation step, thus reducing the culture time and, at the same time, providing the two cell types with respective signals from the beginning. This approach resulted in a simultaneous positive effect on the osteogenic differentiation of MSODs (in terms of gene expression and alizarin red quantification) and on the tubular formation of HUVECs (in terms of branching and vessel density) compared to the monocultures at day 14. Of interest, MSODs were necessary to maintain the HUVEC network at day 14, indicating a potential and not yet documented pericytic function of MSOD analogue to that of MSCs [44]. Finally, the choice of the biomaterial gravitated towards a commercially available methacrylated gelatin due to its easy processability, well-documented biocompatibility, and limited batch-to-batch variability, as confirmed by the uniform distribution and morphological stability of cells throughout the culture period, as well as by the elevated cell viability assessed both in terms of glucose consumption and DNA content.

The engineering of a viable and well-differentiated vascularized bone construct is just the first step towards the modeling of a complex and multifactorial pathology as OA.

The vascularized model proposed here, once exposed to inflammatory factors, was capable of recapitulating most of the OA disease hallmarks, such as bone sclerosis, increased VEGF and ALP activity, and increased catabolic activity and angiogenesis. In particular, gene expression data showed an overall overexpression of *MMP13* and *ADAMTS5* after 1 week of inflammation. Alternatively, the vascularized bone model we used was capable of recapitulating other specific OA features in the presence of the endothelial component. The model showed phenotypes that might be reminiscent of bone sclerosis, such as increased production of collagen type 1, both at the gene and protein level, and demineralization in terms of reduced alizarin red quantification and increased calcium release in the medium. Demineralization of the matrix was linked to the increased Col1α1/Col1α2 ratio found in OA bone due to the fact that the former collagen isoform has reduced ability to retain hydroxyapatite crystals [45]. Our system was able to detect this unbalanced collagen isoforms ratio at the gene expression level, proving to be effective in mimicking specific OA traits in the subchondral bone. Moreover, the co-culture also showed increased levels of ALP activity, whose serum levels were correlated to severity of knee OA in population studies [37]. Interestingly, some OA bone features were already present in the MSOD alone group (e.g., the increased calcium released in the medium and *ADAMTS5* and *MMP13* gene expression), but they were enhanced in the presence of HUVECs. Moreover, other important features, such as the increased *COL1A1*/*COL1A2* gene ratio, the increased ALP activity, and increased OSC, ALP, and VEGF gene expressions, were only observable in the MSOD-HUVECs group, strongly indicating that the presence of endothelial cells in the inflamed environment is crucial to the observation of specific OA phenotypes and could play a role in the onset of those OA traits. At this point, the increased network formation by endothelial cells after inflammation might be reminiscent of the pathological angiogenesis found in OA subchondral bone, which has been identified as a potential marker of early OA [46]. This could be related to the increased expression of the potent angiogenesis stimulator VEGF in the inflamed condition in MSOD/HUVECs co-cultures. Future studies could be performed to assess whether VEGF neutralizing antibodies or VEGF receptors, proven to be effective in alleviating OA symptoms in mice and rats [38], attenuate some of the inflammatory-induced OA traits in our human-based vascularized bone model.

Interestingly, different results were achieved when treating the system with the conditioned medium from mechanically-induced OA engineered cartilage rather than providing the inflammatory cocktail. Specifically, the action of the conditioned medium mostly affected the MSOD component, leading to constructs demineralization and ALP, OSC, and VEGF gene expression alterations. Since inflammation strongly drives angiogenesis through synovial macrophages activation in vivo, it is possible that factors produced by the HPC microcartilage were not sufficient to elicit a response in the vascular component of our model [47]. Overall, a weaker effect was registered if compared to that caused by direct administration of inflammatory factors. This finding could also correlate with the higher OA severity in patients affected by secondary OA due to rheumatoid arthritis, characterized by a preponderant systemic inflammation [48]. Pro-inflammatory factors are mostly produced by chondrocytes, osteoblasts, and synovial cells in primary OA, being detected in the synovial fluid at concentrations in the order of pg/mL [35]. This might explain their absence or very low level in HPC microcartilage condition medium (Appendix A). These factors are amplifiers of angiogenesis [49,50], and their absence in the conditioned medium might explain why HUVEC network formation was not affected and consequently why the co-culture group was not sensitive to the conditioned medium treatment.

Since it has been postulated that cartilage degeneration might be a consequence of aberrant angiogenesis and dysregulated bone function as early events [15,16], it could be speculated that factors produced by cartilage in late phases of the disease might not have a strong effect on the “healthy” engineered vascularized bone in our model. The mechanism by which endothelial cells could contribute to an onset of OA traits needs to be further investigated; however, this model could represent a first step for the study of this complex relationship in the subchondral bone.

## 4. Materials and Methods

### 4.1. Generation of the Constructs and OA Induction

GFP-positive immortalized mesenchymal stromal cells (MSOD cell line: Mesenchymal stromal cells Sword Of Damocles) were generated, as previously described [27], and expanded as a monolayer in Alpha Minimum Essential Medium, containing 10% fetal bovine serum, 2% penicillin/streptomycin/glutamate, 5 ng/mL FGF-2, 1 M HEPES, and 1 mM sodium Pyruvate (growth medium). RFP-positive human umbilical vein endothelial cells (HUVECs) were purchased from Angio-Proteomie (Boston, MA, USA) and expanded as a monolayer in Endothelial Growth Medium 2 (EGM-2, Lonza, Basel, Switzerland)). Both cell types were pelleted and resuspended at a 1:1 ratio in 5% *w*/*v* photocrosslinkable gelatin methacrylate (Cellink) at a density of 20 × 10^6^ cells/mL. In total, 4 μL drops of cell suspension were placed on a 48-well plate and crosslinked for 2 min under UV light. MSODs or HUVECs alone were used as controls. Cells were cultured up to 14 days in a 1:1 ratio of EGM-2 and osteogenic medium [growth medium supplemented with 0.1 mM L-ascorbic acid-2-phosphate (Sigma, St. Louis, MO, USA, A-8960), 0.01 M β-glycerophosphate (Sigma, G9422), and 10 nM dexamethasone (Sigma, D-2915)], then exposed for 1 additional week either to a cocktail of inflammatory cytokines composed of 100 pg/mL IL-6 (R&D 206-IL), 50 pg/mL IL-1β (Sigma, SLBQ7600V), and 50 pg/mL TNFα (Sigma, MKBR4548) or to the HPC microcartilage conditioned medium (1:1 dilution with the culture medium) from cartilage-on-chip exposed to hyperphysiological loading [20].

### 4.2. Cell Viability

The cells’ viability was assessed by Calcein Blue AM staining (Invitrogen, Carlsbad, CA, USA) and by a Glucose Assay kit (Colorimetric/Fluorometric) (Abcam, Cambridge, UK, ab65333), according to the manufacturer’s instructions.

### 4.3. Quantitative RT-PCR Analysis

Sclerotic and non-sclerotic bone pieces were collected from tibial plateaus of resected knees from three OA patients (57 years female, 56 years female, 54 years male), who underwent total joint arthroplasty. Bone pieces of about 100 mg were snap frozen, disintegrated using a FastPrep-24 5G bead beating lysis system (Mpbio, Irvine, CA, USA), and put in Tri Reagent^®^ (Sigma, T-9424). Engineered constructs were sonicated and homogenized in Trizol. Total RNA was then extracted and purified using chloroform, isopropanol/glycogen. and 70% ethanol washes. cDNA was reversely transcribed using the Superscript III kit (Invitrogen). Quantitative real-time PCR was performed using a 7300 AB (Applied Biosystems, Foster City, CA, USA) and TaqMan probes (Applied Biosystems) listed in Table 1. ALP, OSX, and COL1A1 were used to evaluate early osteogenesis; BSP, OPN, and OSC were used to evaluate late osteogenesis [32]; ADAMTS5 and MMP13 were used to evaluate catabolic activity during OA [39]; VEGF, COL1A1/COL1A2, ALP, OSC, and WNT5a were used to evaluate changes in genes specifically modulated during OA [9,12,38]. GAPDH was adopted as a reference gene for the comparative cycle threshold (CT) method.

### 4.4. Alizarin Red Staining and Quantification

Samples were fixed in 4% formalin solution and stained as whole-mount. Alizarin red staining was performed on samples for 2 h at room temperature after conditioning with 1% acetic acid solution for 1 h. Images were taken using a Nikon Eclipse Ti2 microscope (Nikon Instrument, Amsterdam, The Netherlands). Samples were then dissolved again in 1% acetic acid for 1 h, incubated at 90 °C for 5 min, and centrifuged at 12,000× *g* rpm for 15 min. Supernatant was collected and alizarin red positivity was quantified, measuring absorbance at 405 nm in a Synergy H1 Hybrid Multi-Mode Reader (BioTek Instruments, Winooski, VT, USA).

### 4.5. Immunofluorescence

Samples were fixed in 4% formalin solution and either stained as whole-mount or paraffin-embedded and sectioned at 5-μm thickness. Antigens were enzymatically retrieved with incubation in chondroitinase/pronase in 0.02% bovine serum albumin (BSA) solution in phosphate-buffered saline (PBS) for 30 min at 37 °C. Nonspecific binding was then suppressed with 1% horse serum in BSA for 2 h. Following antigen retrieval and blocking, samples were incubated overnight at 4 °C, with primary antibodies against either human collagen type I (Abcam, ab88147), BSP (Abcam, ab3778), or CD31 (Abcam, ab28364) at dilutions of 1:400, 1:100, and 1:50, respectively. Subsequently, samples were incubated with secondary antibodies conjugated with Alexa Fluor 488, 647, and 594. DAPI was used to stain nuclei. Images were acquired with a Nikon Eclipse Ti2 microscope (Nikon Instrument).

### 4.6. Quantification of Alkaline Phosphate (ALP) Activity and Calcium Content in the Surnatants

Supernatant of cultures was collected at specific time points and assayed for ALP activity using the Alkaline Phosphatase Assay Kit (Colorimetric) (Abcam, ab83369) and the Calcium Assay Kit (Colorimetric) (Abcam, ab102505), following the manufacturer instructions. Sample dilutions were made according to values of the standard curves.

### 4.7. Angiogenesis Analysis

Images of whole-mount samples were taken at different timepoints, and HUVECs network formation was measured on 20× magnification pictures using the Angiogenesis analyzer tool of ImageJ program (NIH, Bethesda, MD, USA) [51] on four randomly selected areas of the pictures.

### 4.8. OA Cartilage-on-Chip Model Setup

An in vitro model of OA cartilage-on-chip was introduced, as previously described [20]. Briefly, the device made of hyperelastic polydimethylsiloxane (PDMS) was constituted by two chambers, nominally a culture chamber and a pneumatic chamber, separated by flexible PDMS membrane. The culture chamber consisted of a central gel channel separated by two rows of suspended T-shaped posts from the lateral culture channel for medium supplementation. An interspace was present between the pillar’s bottom surface and the culture’s chamber floor. Upon application of a positive pressure in the actuation chamber (i.e., 0.8 Atm), the flexible PDMS membrane deflected upward until it was stopped by the pillars. Tailoring the relative heights of the gap and the pillars, a defined compression level could be achieved. Devices were designed to provide an hyperphysiological 30% compressive strain, previously demonstrated to be sufficient for the induction of OA traits in cartilage microconstructs. Human articular chondrocytes isolated from healthy donors (two males, 54 years.) were seeded in 3D into microdevices at a density of 5 × 10^4^ cells/mL, lading them in an enzymatically cross-linked and degradable polyethylenglycol-based hydrogel formulation [52]. Chondrocytes were cultured statically in chondrogenic medium for 2 weeks [20], demonstrated to be sufficient for the development of cartilaginous-like constructs rich in COL2A1 and ACAN. Constructs were then mechanically stimulated for 7 further days with a previously adopted regimen (1 Hz, 2 cycles of 2 h per day, with a 4 h pause in between) through a custom-made pressure controller [20]. Constructs cultured statically for 3 weeks were adopted as controls. Culture medium was replenished every other day and collected for conditioning of MSPD-HUVEC constructs and analysis, as described below.

### 4.9. Microfluidic Devices Fabrication

Devices were fabricated in PDMS (Sylgard 184) through classic photolithography and replica molding techniques. Device master molds were realized through multi-layer photolithography of SU-8 photoresist (Microchem) onto 4 inches silicon wafers, realized in a cleanroom environment. PDMS layers were realized with defined casts, with a 10:1 weight ratio of polymer precursor to curing agent and left to polymerize at 65 °C for at least 3 h. PDMS stamps were then carefully peeled off molds and the appropriate inlets for the hydrogel compartments (diameter 1 mm), the actuation compartment (diameter 1.5 mm), and culture medium reservoirs (diameter 5 mm) bored with biopsy punchers. Final devices were assembled through layers of air plasma activation (Harrick Plasma, Ithaca, NY, USA) and further incubation for 30 min at 80 °C to achieve irreversible adhesion after conformal contact.

### 4.10. Supernatant Analyses

Undiluted conditioned media from HPC microcartilage were assayed to detect the presence of inflammatory cytokines. IL-1β, IL-6, and TNFα were analyzed by a Luminex^®^ Assay using the Human Premixed Multi-Analyte Kit (R&D Systems, Minneapolis, MN, USA), following the manufacturer’s instructions, and readings were acquired using a BioPlex MagPlex Beads (Magnetic) system (Bio-Rad, Hercules, CA, USA). IL-8 concentration was measured using the Human IL-8 ELISA Set (BD Biosciences), following the manufacturer’s instructions, and readings were acquired using a Synergy H1 Hybrid Multi-Mode Reader (BioTek Instruments).

### 4.11. Statistical Analysis

Experiments were conducted in triplicate, and quantitative results were reported as mean ± standard deviation. Statistical analysis was performed using GraphPad Prism 9. For normally distributed samples, paired comparisons were analyzed using a two-tailed *t*-test, whereas one-way ANOVA and the post-hoc Tukey test were performed for multiple comparisons. If population did not pass the normality test, the Whitney-Man test was performed to compare paired samples, while the Kruskal-Wallis and Dunn tests were performed for multiple comparisons. Number of * indicates level of significance in the pictures. ns = non-significant, * = *p* < 0.05, ** = *p* < 0.005, *** = *p* < 0.0005, **** *p* < 0.00005.

## 5. Conclusions

The vascularized bone model we used, once exposed to inflammatory or HPC microcartilage derived factors, was capable of recapitulating some early OA tissue features, such as key gene expression changes and ALP activity, as well demineralization and increased collagen production reminiscent of bone sclerosis. In particular, network formation by endothelial cells was increased in the case of inflammation, reminiscent of the abnormal angiogenesis found in OA subchondral bone. Interestingly, some features were already present in the MSOD alone group but were enhanced in the presence of HUVECs, while others were only observable in the MSOD-HUVECs group, indicating that the presence of endothelial cells in the inflamed environment could play a role in the onset of OA traits. The mechanism by which endothelial cells could contribute to OA traits onset still needs to be elucidated; however, this model could represent a first step for the study of this complex relationship. In a broader perspective, our vascularized bone model could also be directly combined with a cartilage layer to study cartilage/bone changes in vitro under OA mimicking conditions so as to possibly highlight new pathological mechanisms but also screen anti-OA drugs.

## Figures and Tables

**Figure 1 ijms-22-09581-f001:**
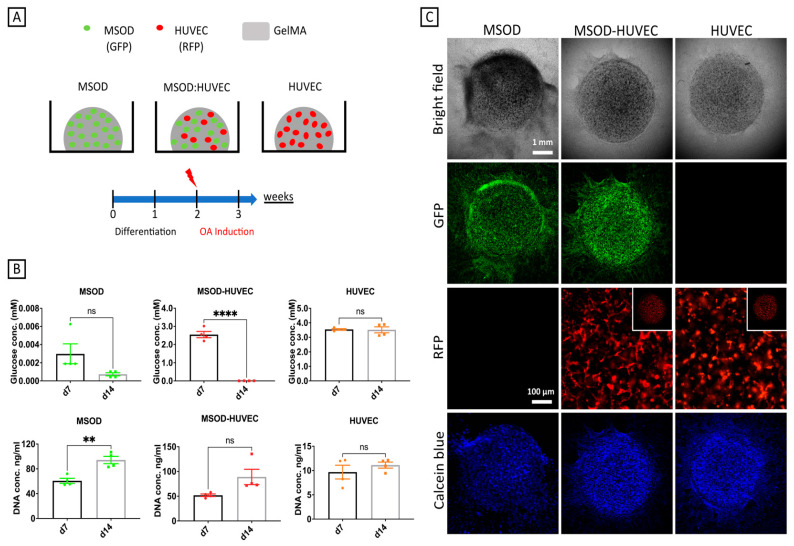
Generation of the vascularized bone model. (**A**) Schematic of the culture conditions and experimental timeline. (**B**) Viability analyses of the constructs at 7 and 14 days of culture, showing glucose consumption and DNA content. (**C**) Morphological appearances of constructs after 14 days of culture, showing viable cells positive for green fluorescent protein GFP (MSODs), red fluorescent protein RFP (HUVECs), and Calcein Blue AM (MSOD and HUVEC). Number of experiments = 3, number of replicates/experiment = 4. ** *p* < 0.005, **** *p* < 0.00005.

**Figure 2 ijms-22-09581-f002:**
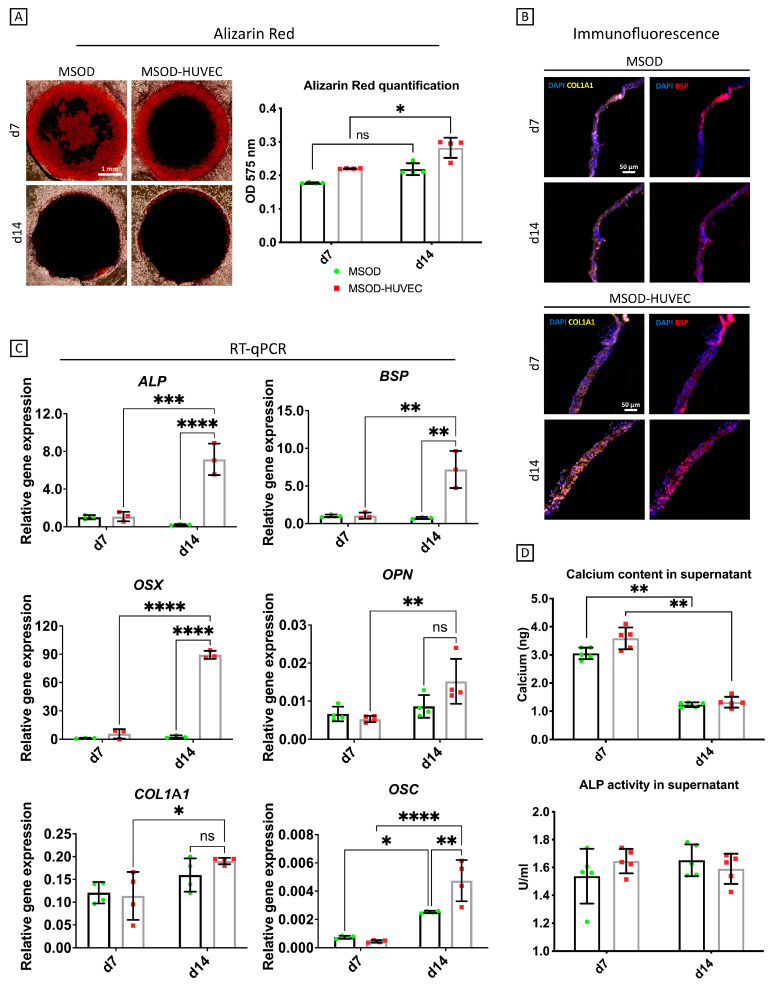
Osteogenic characterization of the vascularized bone model. (**A**) Whole mount alizarin red staining imaging and quantification after digestion of the constructs. (**B**) COL1A1 and BSP immunofluorescence characterization of the cross-section of the constructs. (**C**) Quantitative RT-qPCR analyses; gene expression is relative to GAPDH levels. ALP = alkaline phosphatase, BSP = bone sialoprotein, OSX = osterix, OPN = osteopontin, COL1A1 = collagen type I, OSC = osteocalcin. (**D**) Quantitative evaluation of osteogenesis in terms of calcium released in the medium and ALP activity in the culture supernatant. Number of experiments = 3, number of replicates/experiment = 4. ns = non-significant, * = *p* < 0.05, ** = *p* < 0.005, *** = *p* < 0.0005, **** *p* < 0.00005.

**Figure 3 ijms-22-09581-f003:**
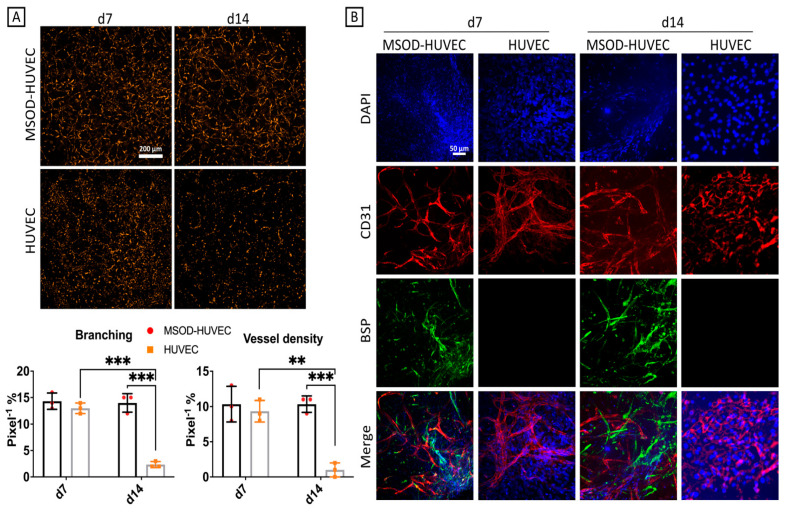
Tubulogenesis characterization of the vascularized bone model. (**A**) Live RFP fluorescence imaging and quantification of the RFP-HUVEC network formation alone and in co-culture, showing more tubulogenesis in co-culture with MSOD. (**B**) Immunofluorescence imaging of CD31 and BSP. Number of experiments = 3, number of replicates = 4. ns = non-significant, ** = *p* < 0.005, *** = *p* < 0.0005.

**Figure 4 ijms-22-09581-f004:**
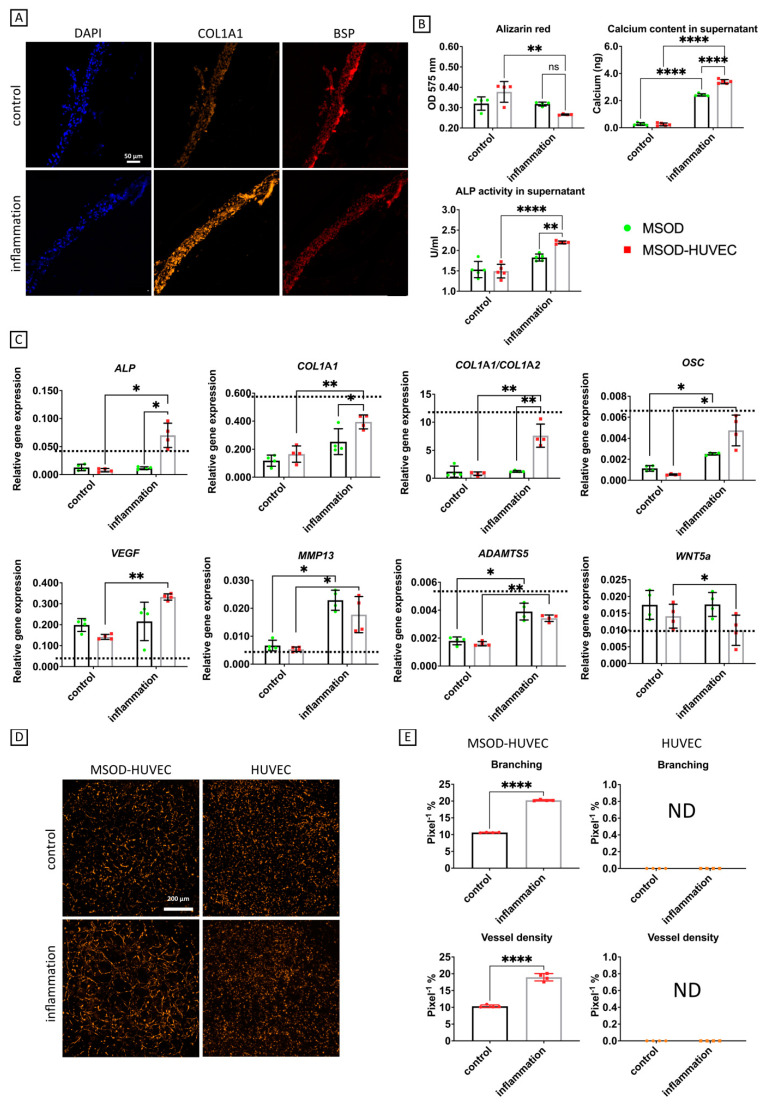
OA induction through an inflammatory cocktail in the vascularized bone model. (**A**) Immunofluorescence characterization of COL1A1 and BSP. (**B**) Biochemical characterizations. (**C**) Quantitative RT-qPCR analyses; gene expression is relative to GAPDH levels. Dashed lines represent gene expression values of human sclerotic bone isolated from OA patients who underwent total joint arthroplasty (*n* = 3). ALP = alkaline phosphatase, COL1A1 = collagen type I, OSC = osteocalcin, VEGF = vascular endothelial growth factor, MMP13 = matrix metallopeptidase 13, ADAMTS5 = A disintegrin and metalloproteinase with thrombospondin motifs 5. (**D**,**E**) Characterization of network formation by HUVEC in OA induction by imaging and angiogenesis quantification, respectively. Number of experiments = 3, number of replicates/experiment = 5. ns = non-significant, * = *p* < 0.05, ** = *p* < 0.005, **** *p* < 0.00005.

**Figure 5 ijms-22-09581-f005:**
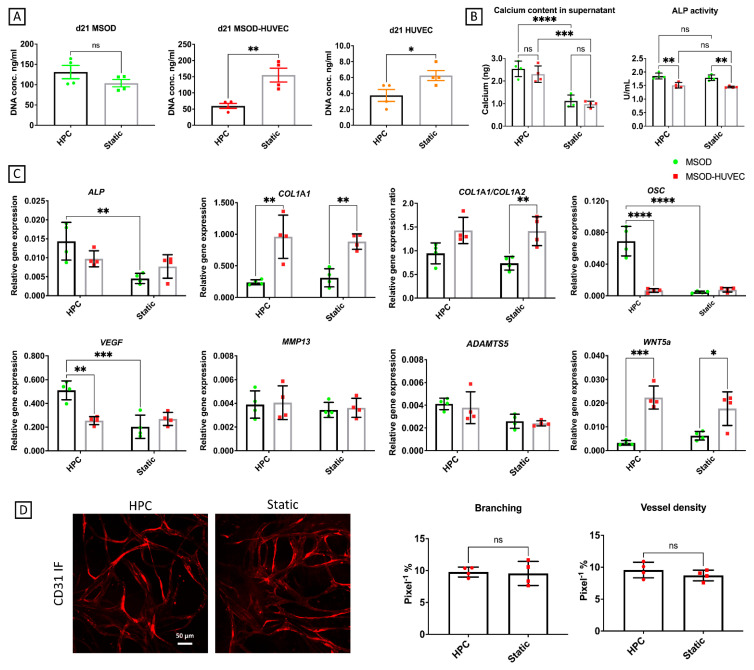
Effects of the conditioned medium from HPC microcartilage on the vascularized bone model. (**A**) DNA content analysis to measure cell viability. (**B**) Biochemical characterizations of the culture supernatant. (**C**) Quantitative RT-PCR analyses; gene expression is relative to GAPDH levels. ALP = alkaline phosphatase, COL1A1 = collagen type I, OSC = osteocalcin, VEGF = vascular endothelial growth factor, MMP13 = matrix metallopeptidase 13, ADAMTS5 = A disintegrin and metalloproteinase with thrombospondin motifs 5. (**D**) Characterization of network formation by HUVECs in the MSOD-HUVEC group by imaging and angiogenesis quantification, respectively. Number of experiments = 1, number of replicates/experiment = 4. ns = non-significant, * = *p* < 0.05, ** = *p* < 0.005, *** = *p* < 0.0005, **** *p* < 0.00005. HPC = HPC microcartilage conditioned medium, Static = conditioned medium from microcartilage not mechanically stimulated.

**Table 1 ijms-22-09581-t001:** List of probes used for gene expression.

Gene	Assay on Demand Ref. No.
*GAPDH*	Hs02758991_g1
*ALP*	Hs01029144_m1
*OSX*	Hs00541729_m1
*BSP*	Hs00173720_m1
*OPN*	Hs00959010_m1
*COL1A1*	Hs01097664_m1
*COL1A2*	Hs01028960_m1
*OSC*	Hs01587814_g1
*VEGF*	Hs00900055_m1
*WNT5a*	Hs00998537_m1
*ADAMTS5*	Hs00199841_m1
*MMP-13*	Hs00233992_m1

## Data Availability

All data are available in the present article and Appendix A.

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
