# Peer review of "Modeling In Vitro Osteoarthritis Phenotypes in a Vascularized Bone Model Based on a Bone-Marrow Derived Mesenchymal Cell Line and Endothelial Cells"

_ijms, 2021, doi:10.3390/ijms22179581_

Round 1

Reviewer 1 Report

the authors should remove colloquialisms such as "on the other hand" and "to the best of the author's knowledge", "In this work". Furthermore, scientific writing should exclusively be from the third person perspective. Remove all first person references.

the data and the conclusions all appear reasonably well justified 

Author Response

The authors removed formalities and revised the text according to the reviewer’s suggestions.

Reviewer 2 Report

In this paper, the authors describe an interesting model of vascularised human bone, using a co-culture of an in-house developed MSC line (MSOD) with HUVECs. The authors utilise the model to good effect to try and elucidate the reaction of the model to various conditions mimicking aspects of OA. The development of a simple, reproducible, vascularised bone model is of great interest to me personally and to the wider research community. While the experimental design and outcomes are appropriate, there are areas, mainly relating to the framing of the study and the presentation of the data, that could be improved.

Introduction

  1. Line 108: Add a few words to clarify the aim of the study – e.g. ‘…therefore validating the system as a simple, but biologically-relevant, model of OA bone for use in future research’.
  2. Line 91-94: The mention of microfluidic technologies in the introduction is confusing and largely unnecessary. The authors describe that microfluidics have ‘allowed the development of vascularised and even innervated bone-on-chip models’ but that these have not been used to study OA bone. This makes it sound as though the model presented in this study will therefore use microfluidics to this effect, but no microfluidics are involved in the MSOD/HUVEC co-culture. Consider removing reference to microfluidics.
  3. Line 78: Indent missing.
  4. It is not clear in the introduction that encapsulating HUVECs in a hydrogel with tissue-specific cells is an established method for generating mimics of vascularised tissues. Consider adding a section referencing other studies that have used this method for similar applications e.g. Liu, J., Chuah, Y., Fu, J., Zhu, W. and Wang, D., 2019. Co-culture of human umbilical vein endothelial cells and human bone marrow stromal cells into a micro-cavitary gelatin-methacrylate hydrogel system to enhance angiogenesis. Materials Science and Engineering: C, 102, pp.906-916.

Results

  1. Line 110 and 112: ‘gelMA’ – abbreviations need to be written in full the first time they appear.
  2. Figure 1B: Graphs are too small to read. Consider increasing their size.
  3. Figure 1: GFP and RFP – abbreviations not explained in full.
  4. Figure 2A and 2C: Graphs are too small, axes are impossible to read. Consider making figures full page.
  5. Figure 3A: Again, graphs are far too small to make out.
  6. Figure 4: Again, graphs are very small and the axes are difficult to make out.
  7. Line 214: Gene names throughout should be in italics. Gene name is COL1A1 not COL1α1.
  8. Figure 5: Again, graphs are too small and difficult to make out.
  9. Line 199, 218, 230: Figure 4 references have a lower-case letter.
  10. Line 241: ‘Previously demonstrated…’ needs a reference.
  11. Line 242: space missing: ‘microfluidicdevices’
  12. Line 244: Cartilaginous misspelt.

Discussion

  1. Line 321-330: This section would be better in the introduction (line 52) as it introduces the biological problem. Then briefly touch upon in the discussion.
  2. Line 371: Cartilage degeneration is not exclusively a sign of late-OA. The cartilage degeneration that is modelled in your study using OA chondrocytes and hyper-physiological loading is representative of late-OA. Consider re-phrasing.
  3. Line 361-370: It would be nice to relate findings back to clinical relevance, linking to the different subtypes/causes of OA.
  4. Line 368: Add in some discussion as to why you don’t think there were inflammatory cytokines in the conditioned media.

Materials and Methods

  1. There is no mention at all of the set-up of the ‘OA cartilage-on-a-chip’. Please add a section to the materials and methods describing the nature of this model. What is the cell source? How is the model set-up? At the very least, there needs to be a short description in the materials and methods and a reference to Occhetta et al., 2019 [16].

Author Response

The authors thank the reviewer for the conctructive comments and modified text and figures according to the reviewer’s suggestions.

Introduction

  1. The authors added more details about the work aim.
  2. The authors removed the microfluidics paragraph in the introduction.
  3. The autors added the missing indent.
  4. The authors added this section referencing other studies on co-culturing endothelial cells and tussue-specific cells.

Results

  1. The authors added abbreviations.
  2. The authors increased the graphs’ size.
  3. The autors explained the abbreviations.
  4. The authors increased the graphs’ size.
  5. The authors increased the graphs’ size.
  6. The authors increased the graphs’ size.
  7. The authors corrected the genes’ font and nomenclature.
  8. The authors increased the graphs’ size.
  9. The authors corrected Figure 4 reference.
  10. The authors added the reference.
  11. The authors added the space.
  12. The authors corrected this spelling.

Discussion 

  1. The author transferrred the section in the introduciton and mentioned that in the discussion as suggested.
  2. The authors rephrased that sentence.
  3. The authors modified the discussion accordingly.
  4. The authors expanded this part of the discussion.

Materials and Methods

The authors added this section to Material and Methods

Round 2

Reviewer 1 Report

the manuscript has not been revised to the third person voice

Author Response

We have now revised our manuscript to third person voice

Round 3

Reviewer 1 Report

 I am now satisfied